# The Effects of BSA-Stabilized Selenium Nanoparticles and Sodium Selenite Supplementation on the Structure, Oxidative Stress Parameters and Selenium Redox Biology in Rat Placenta

**DOI:** 10.3390/ijms232113068

**Published:** 2022-10-28

**Authors:** Milica Manojlović-Stojanoski, Slavica Borković-Mitić, Nataša Nestorović, Nataša Ristić, Svetlana Trifunović, Magdalena Stevanović, Nenad Filipović, Aleksandar Stojsavljević, Slađan Pavlović

**Affiliations:** 1Institute for Biological Research “Siniša Stanković”—National Institute of the Republic of Serbia, University of Belgrade, Bulevar Despota Stefana 142, 11060 Belgrade, Serbia; 2Group for Biomedical Engineering and Nanobiotechnology, Institute of Technical Sciences of the Serbian Academy of Sciences and Arts (SASA), Kneza Mihaila 35/IV, 11000 Belgrade, Serbia; 3Faculty of Chemistry, University of Belgrade, Studentski trg 12-16, 11000 Belgrade, Serbia; 4Innovative Centre, Faculty of Chemistry, University of Belgrade, Studentski trg 12-16, 11000 Belgrade, Serbia

**Keywords:** selenium, placenta, rats, oxidative stress parameters, nanoselenium, embryo, fetus

## Abstract

The chemical element selenium (Se) is a nonmetal that is in trace amounts indispensable for normal cellular functioning. During pregnancy, a low Se status can increase the risk of oxidative stress. However, elevated concentrations of Se in the body can also cause oxidative stress. This study aimed to compare the effects of BSA-stabilized Se nanoparticles (SeNPs, Se^0^) (BSA-bovine serum albumin) and inorganic sodium selenite (NaSe, Se^+4^) supplementation on the histological structure of the placenta, oxidative stress parameters and the total placental Se concentration of Wistar rats during pregnancy. Pregnant females were randomized into four groups: (i) intact controls; (ii) controls that were dosed by daily oral gavage with 8.6% bovine serum albumin (BSA) and 0.125 M vit C; (iii) the SeNP group that was administered 0.5 mg of SeNPs stabilized with 8.6% BSA and 0.125 M vit C/kg bw/day by oral gavage dosing; (iv) the NaSe group, gavage dosed with 0.5 mg Na_2_SeO_3_/kg bw/day. The treatment of pregnant females started on gestational day one, lasted until day 20, and on day 21 of gestation, the fetuses with the placenta were removed from the uterus. Our findings show that the mode of action of equivalent concentrations of Se in SeNPs and NaSe depended on its redox state and chemical structure. Administration of SeNPs (Se^0^) increased fetal lethality and induced changes in the antioxidative defense parameters in the placenta. The accumulation of Se in the placenta was highest in SeNP-treated animals. All obtained data indicate an increased bioavailability of Se in its organic nano form and Se^0^ redox state in comparison to its inorganic sodium selenite form and Se^+4^ redox state.

## 1. Introduction

Selenium (Se) is a micronutrient with a variety of roles which are an essential trace element required for proper physiological functioning [1]. The human genome contains 25 genes encoding selenoproteins which are involved mainly in redox homeostasis [2]. A small amount of Se is vital for antioxidative defense, thyroid hormone production, metabolism, DNA synthesis, fertility, immunity and cellular redox balance [3]. Small differences in Se concentration in the organism determine whether it is toxic or its potential deficiency, and its optimal influence on physiological processes [4]. It is well known that the Se response curve is U-shaped, and that high concentration of Se can have a detrimental effect [5]. Since Se deficiency affects up to a billion people worldwide, particularly in Asian countries, mostly due to inadequate dietary intake, an ideal Se formulation is required to increase its concentration in the body and ensure the beneficial action of selenoproteins [6].

Selenium is crucial in the initial stages of pregnancy, and its deficiency is reflected in developmental processes [7,8]. Some authors have indicated that a low maternal Se status is related to miscarriage [7,9,10], preeclampsia, pregnancy-induced hypertension [11], preterm birth [12] and small-for-gestational-age newborns [13]. Selenium deficiency is present in a third of healthy pregnant women and since it is transferred from the mother to the fetus, its supplementation can have a preventive effect on multiple complications in both mother and fetus [12]. Additionally, Se deficiency in pregnancy affects the functioning of the mother’s thyroid gland, which can indirectly cause fetal growth restriction [13]. The complex role of Se during pregnancy is not fully elucidated and there are no uniform recommendations and guidelines for its use [14]. Considerable effort has been invested in the field of nanotechnology to determine a form of Se with high bioavailability and low toxicity [15]. It would appear that the most promising form is Se in nanoparticles (SeNPs), which has already been shown to be less toxic and have higher bioavailability than the currently widely used inorganic sodium selenite (NaSe) and organic selenomethionine forms, mostly because the Se is in a zero oxidation state (Se^0^) [16].

Selenium is an essential cofactor for many selenoproteins, such as glutathione peroxidases (GSH-Px1, GSH-Px2, GSH-Px3, GSH-Px4 and GSH-Px6), thioredoxin reductases (TXNRD1, TXNRD2, and TXNRD3), iodothyronine deiodinases (DIO1, DIO2 and DIO3), methionine-*R*-sulfoxide reductase 1 (MSRB1) and selenophosphate synthetase 2 (SEPHS2) [2]. Selenoproteins without known functions are assigned with symbols SEL or SEP [2]. All of them are involved in scavenging free radicals and maintaining redox balance [17], i.e., the balance between the production of reactive oxygen (ROS) and reactive nitrogen species (RNS) and their elimination by protective mechanisms. All tissues and organs synthesize selenoproteins, but the level of their synthesis depends on concentration of Se in certain organs [18]. Many studies have demonstrated that Se supplementation can lower oxidative stress induced by different triggers. However, Se compounds can also generate ROS through different pathways and Se toxicity occurs when oxidation exceeds the activity of the antioxidant system [19]. Selenium in its nanoform (SeNPs) has been shown to provoke oxidative defense against ROS and RNS, while in high doses it causes oxidative stress, depending on the redox environment in cells [20,21].

The placenta is a transient structure comprising fetal membranes and uterine mucosa; its function is to support a developing fetus [22]. The placenta has an active role in Se transport during the entire course of intrauterine development [18]. A ligand–receptor mechanism mediates the endocytosis of the transport protein selenoprotein P from the maternal circulation, allowing Se transport against a concentration gradient to the fetus [21]. When placentation (remodeling of spiral arteries by extravillous trophoblasts) is inadequate, the resulting suboptimal and deficient placental function leads to complications in pregnancy and subsequently to fetal growth restriction due to hypoxia [23]. 

The rise in oxygen tension in the intervillous space is a physiological phenomenon in normal placentation that can lead to placental oxidative stress [24]. To compensate for oxidative stress, the placenta increases its antioxidant activity [24]. One of the reasons for oxidative stress is the deterioration of syncytiotrophoblasts in the presence of high oxygen levels, resulting in cell vacuolization, a reduction in surface microvilli and a decline in the number of mitochondria without ensuing damage to cytotrophoblasts and stromal cells [25]. If the response to oxidative stress is inadequate, the insufficient increase in antioxidant capacity leads to changes in mitochondrial ultrastructure and moderate syncytiotrophoblast damage [26], resulting in a state of chronic oxidative stress. Another hypothesis explains the oxidative stress in the placenta by intermittent maternal blood flow in the intervillous space, which results in ischemia-reperfusion injury (IRI) that is also mediated through the generation of ROS via different pathways, for example the mitochondrial electron transfer processes and activity of different enzymes, including NADPH oxidase [27].

Se dosage is of crucial importance during supplementation in pregnancy [28]. Developing organisms are the most vulnerable and the resulting changes can have permanent and lifelong health consequences. Therefore, the assessment of fetotoxicity after the administration of nanoparticles during pregnancy is very important. [29,30]. Similarly, exposure to diverse types of nanoparticles during pregnancy (silica, gold or titanium dioxide nanoparticles, and carbon nanotubes) resulted in impaired placental functioning, disrupted fetal development, and increased in utero mortality [31,32,33,34] 

The aim of this study was to compare the effects of two different forms of Se: BSA-stabilized Se nanoparticles (Se^0^) and inorganic sodium selenite (Se^+4^), on the structure, oxidative stress parameters and the accumulation of Se in the placenta of pregnant rats. In our experiments, we used the same supranutritional and non-toxic concentrations of both forms of Se [35,36]. For the determination of the cellular redox status, the following oxidative stress parameters were investigated in the placenta: superoxide-dismutase (SOD, EC 1.15.1.1), catalase (CAT, EC 1.11.1.6), glutathione peroxidase (GSH-Px, EC 1.11.1.9), glutathione reductase (GR, EC 1.6.4.2), phase II biotransformation enzyme glutathione S-transferase (GST, EC 2.5.1.18), as well as nonenzymatic components, the total glutathione (GSH) content and the concentration of sulfhydryl groups (SH).

## 2. Results

### 2.1. Effects on Pregnant Females—General Parameters

The body weight of pregnant females on the 21st day of pregnancy was significantly decreased by the treatment with SeNPs, while it was not altered by NaSe, (*p* < 0.05) (Appendix A) when compared to the IC and C groups.

In all pregnant females from IC, C, and NaSe groups on the 21st day of pregnancy, the fetuses were viable. However, in the SeNP group, miscarriages were detected in 42% of pregnant females (5/12). In 17% (2/12) of sperm-positive females, embryonal resorptions were detected; they were identified as pinpoint hemorrhagic placentation sites in uterine horns that were asymmetrically distributed (Figure 1A,B). Subsequent histological analysis revealed that these pinpoint hemorrhagic sites corresponded to maternal hemorrhage in caverns filled with fibrinoid material and blood. These features of uterine walls were signs of embryonal death, disintegration and assimilation (Figure 1C).

In 25% of dams (3/12), late miscarriages were observed as aborted fetuses and identified in the uteri on the day of sacrifice. On the histological sections (Figure 2), signs of pregnancy progression were evident by the presence of chorionic villi that were branched with a loose mesenchyme core covered by thin layers of trophoblasts. In the uterine lumen, a necrotic hyaline content surrounded by an amniotic membrane was observed as the remnants of a dead fetus.

Uterine architecture in early and late miscarriages was considerably different. In early miscarriages, the uterine wall was thin and without crypts. The lumen was surrounded by a single layer of cylindrical epithelium. Stromal endometrial glands were small in both size and number. The myometrium consisted of layers of inner longitudinal and outer circular smooth muscle cells with the tunica vasculosa surrounded by the perimetrium (Figure 3A–C). In late miscarriages, the uteri exhibited all characteristics of gravidity. The endometrium was hyperemic and swollen, highly folded, with crypts and microvascular developments. The luminal epithelium was hypertrophied and pseudostratified. Endometrial stroma was filled with endometrial glands of different sizes that were lined with squamous, cuboidal or cylindrical epithelial cells. In the myometrium, the tunica vasculosa was pronounced (Figure 3D–F).

### 2.2. Effects on the Fetoplacental Unit

A significant decrease in the weights of fetuses of SeNP-treated mothers was observed when compared to the fetal weight of control mothers. In contrast, a significant increase (*p* ˂ 0.005) of this parameter was observed in fetuses of NaSe-treated mothers. The placental weight and the relative placental weight, expressed as the viscera index of the placenta, remained unchanged after the treatments (Table 1). There were no statistically significant differences between sexes in basic fetoplacental parameters on day 21 of pregnancy.

### 2.3. Histological and Stereological Analysis of the Placenta of 21-Day-Old Fetuses

The rat placenta of 21-day-old fetuses was comprised of three distinguishable layers: the labyrinth, basal zone and decidua basalis. The labyrinth is the dominant part of the placenta, comprised of maternal sinusoids, trophoblastic septa and fetal capillaries. The basal zone was located below the labyrinth and was comprised of spongiotrophoblasts and trophoblastic giant cells, while glycogen cells were degenerated at this stage of pregnancy. The decidua basalis which is located below the basal zone, is the maternal part of the placenta, and at the end of pregnancy it regressed and included tightly arranged decidual cells and blood vessels. The yolk sac, a residual structure of the fetal membrane and metrial glands of the uterine part were also present (Figure 4).

Treatments did not change the histological structure of the placenta nor its absolute volume as determined stereologically (Appendix A). No changes were observed between their volume densities and the ratio of the placental layers to the entire placenta. Sex-specific differences were not observed in any of the analyzed parameters.

### 2.4. Oxidative Stress Parameters in the Placenta

Examination of the parameters of oxidative stress in both female and male fetuses showed an increase in the activities of CAT and GSH-Px in the placenta of animals that were treated with SeNPs when compared with the IC group and in SOD and GST activities (*p* < 0.05) when compared with the control group. The concentrations of GSH and SH in the SeNP group were significantly decreased when compared to the IC group (*p* < 0.05). In the placenta of rats treated with NaSe, the activity of CAT was significantly lower than in the IC, C and SeNP groups (*p* < 0.05). The concentration of GSH was increased in the placenta of NaSe animals when compared to IC, C and SeNP animals (*p* < 0.05) (Table 2).

The effect of the treatment on all investigated variables separately in females and males is presented in Figure 5A and Table 3. There was a significant increase in SOD and GSH-Px activities in the SeNP vs. the C group in the placenta of female fetuses, and a significant decrease in GST activity was observed between the SeNP and IC groups in both sexes (*p* < 0.05).

In placentas of the SeNP group of male fetuses, we detected an increase in the activities of SOD, CAT and GSH-Px, and a decrease in the GSH concentration compared to the IC group (*p* < 0.05). In the SeNP group we observed significant increases in placental SOD, GSH-Px, GR and GST activities when compared to the C group (*p* < 0.05).

In the placenta of female fetuses of the NaSe group, we observed a decrease in GST activity and an increase in GSH concentration when compared to all other investigated groups of animals, as well as a decrease in SH concentration in respect to the IC group (*p* < 0.05). In the placenta of male fetuses of the group treated with NaSe, GST activity was significantly reduced compared to males of the IC and SeNP groups (*p* < 0.05). The concentration of GSH in the placenta of male fetuses of the NaSe group was significantly increased in comparison to the C and SeNP groups (*p* < 0.05).

After taking into account the differences between the placentas of female and male fetuses (e.g., the effects of gender), we observed that the activity of GSH-Px was significantly higher in the placenta of male fetuses in the C group (*p* < 0.05) and that the activity of GR was significantly increased in placental samples of male fetuses when compared to female fetuses in the IC group (Figure 5B; Table 3).

The results of principal component analysis (PCA) are presented in Figure 6 and Table 4. PCA was employed to examine the possible discrimination of all examined groups (treatments) based on all investigated antioxidant parameters. PCA was performed in two ways: by projecting the relative contribution of every antioxidant component in the factor plane (Figure 6A), and by projecting the investigated groups based on the antioxidant defense parameters (Figure 6B). Table 4 shows the loadings of the variables onto the principal components (PCs). The PCA of the relative contribution of every antioxidant component showed that PC1 and PC2 (Figure 6A) can explain about 50% of the total variance in the data matrix. PC1 explains 31.74% of the total variance, with SOD, CAT and GSH-Px as the parameters that contributed the most to the discrimination. PC2 explains 17.50% of the total variance with SH groups, with CAT and GSH-Px as the parameters that contributed most to the separation. A summary of the PCA results for all investigated groups of animals considering all parameters of oxidative stress (Figure 6B) indicates that PC1 and PC2 can explain about 90% of the total variance. PC1 (55.82%) distinguished IC and C groups from SeNP and NaSe groups, while PC2 (30.07%) distinguished the NaSe group from all other examined groups.

### 2.5. Level of Se in the Placenta

The concentration of placental Se is presented in Table 5. We compared the recorded results in three ways as follows: females + males, only females and only males. The concentration of Se was significantly increased in all groups treated with Se, SeNPs and NaSe compared to the IC and C groups (*p* < 0.05). The concentration of Se was also found to be more significantly elevated in the C group than in IC. Most importantly, in placental samples from the SeNP group, the concentration of Se was 2-fold higher than in placental samples from the NaSe-treated group (*p* < 0.05).

## 3. Discussion

The results presented herein show that, unlike in adulthood, the toxic effects of exposure to SeNPs manifest during fetal development; because of the SeNP application during pregnancy, miscarriage was observed in 42% of pregnant females. In pregnant females with viable fetuses, the treatment with SeNPs affected the placental parameters of oxidative stress showing that functional changes were present at the level of placental antioxidant defense due to increased placental Se levels on day 21 of gestation. The hostile gestational environment caused by maternal SeNP intake was further reflected in the decreased weight of near-term fetuses. In parallel, maternal NaSe treatment affected fetal parameters in the opposite direction: increased fetal body weight and slightly increased placental Se resulting in healthy-looking viable fetuses which were present in all pregnant females of the studied experimental group. Neither Se treatments provoked any structural changes in the placenta.

The RDA (recommended daily allowance) of Se for both men and women is 55 µg/day. The Tolerable Upper Intake Level (UL) for adults is 400 µg/day. Selenium deficiency affects the Se-dependent components of the antioxidant defense system. Selenium supplementation can increase the activities of Se-dependent proteins/enzymes [37]. However, at levels above the UL, Se compounds can be toxic. Until now, Se has been used as a supplement in both its organic and inorganic forms. Selenium nanoparticles (SeNPs) present a novel approach to nutrition supplementation, while in medicine they are a promising agent with a wide use for diagnostic and therapeutic purposes [38,39]. A growing body of literature data demonstrates that stabilization of Se in nanoparticles in its zero oxidation state reduces its toxicity in relation to other Se forms, enhances its bioavailability and free radical scavenging efficiency and increases its therapeutic potential [29,40]. Therefore, the SeNPs used in our study are defined as highly water-soluble nano-red elemental Se particles of a nano-defined size (≈50 nm) in a zero redox state (Se^0^) and applied in a dose of 0.5 mg SeNPs/kg/bw. There have been varying opinions regarding what is an optimal and what is a supranutritional, non-toxic dose of Se in rats [35,36]. According to the Rayman (2020) [41], tolerance toward Se in humans depends on genes that regulate their ability to tolerate low or high doses of Se, but by now this has not been fully elucidated. Optimal dose of Se varies between populations and individuals [41]. However, the chosen dose of synthesized SeNPs (0.5 mg/kg bw) was shown to be supranutritional and non-toxic, based on testing of adult male rats [35,36]. The results of chronic exposure to SeNPs in the range of 0.2–0.8 mg/kg bw did not exhibit any toxic effect, exhibiting no histopathological changes of the liver, kidneys, lungs and blood biochemistry parameters [42]. In contrast to the results obtained in adult males, SeNP dosing by oral gavage during pregnancy negatively affected prenatal development and caused embryo-fetal lethality. Similarly, exposure to diverse types of nanoparticles during pregnancy (silica, gold or titanium dioxide nanoparticles, and carbon nanotubes) impaired placental functioning, disrupted fetal development and increased in utero mortality [31,32,33,36].

Treatment with SeNPs impeded preimplantation embryonic development and adversely affected the process of embryo implantation, causing asymmetric implantations with miscarriage as an outcome in 17% of the examined pregnant females. From the very beginning, the developing embryo adjusts its energy metabolism and antioxidative defense system in accordance to varying environmental conditions regarding the availability of oxygen and glucose in different areas of the maternal reproductive system [43]. Maternal dosing with SeNPs by gavage from the first day of pregnancy led to an increase in Se levels to which preimplantation embryos were exposed. Due to the spherical morphology of SeNPs with a diameter of ≈50 nm, they are efficiently absorbed by the intestinal mucosa, and via the circulation they reach the uterus and uterine fluid [15]. Therefore, exposure to an increased amount of nanoparticles could affect the expression/activity of members of the antioxidative defense system before and after embryo implantation, particularly the Se-dependent GSH-Px enzyme, which is included in defenses against H_2_O_2_ and organic hydroperoxides [15,29,44]. Pinpoint hemorrhagic placentation sites demonstrated that miscarriage was the ultimate consequence since immature homeostatic mechanisms were incapable of controlling the balance between ROS production and elimination [45,46], highlighting the embryonic period as a timeframe of sensitivity. Furthermore, feticide was recorded in 25% of pregnant females that were SeNP-fed, also pointing to disturbances at the level of the antioxidative defense system as one of the crucial mechanisms of the pleiotropic actions of SeNPs [31]. Damage caused by ROS overexposure can include lipid peroxidation, DNA impairment and protein alteration, leading to cell apoptosis with a fatal in utero outcome [47]. Consequently, the period after the formation of the placenta is also critical for external influences, as the action of SeNPs was targeted at the placenta itself and markedly affected the development process. However, in the NaSe-treated group, miscarriages were not observed.

Oral intake is a common route of Se intake in humans, so oral gavage of pregnant rats allowed the particles to pass through the same barriers during their translocation from mother to fetus. It has been shown that Se from nanoparticles is more soluble and bioavailable compared to inorganic forms [48]. Consequently, our results clearly show that the amount of accumulated Se in the placenta after the SeNPs administration was doubled in relation to the group receiving NaSe, while a 4-fold increase was measured compared to the control values. The recorded Se concentrations in placentas exposed to SeNPs are in the range of toxic doses that lead to an oxidoreductive imbalance, which was further reflected in growth retardation and decreased weight in 21-day-old fetuses. In contrast, fetuses of NaSe-treated mothers were viable, with increased fetal weight. Although parameters outside the control range could be challenging, an increase in the weight of near-term fetuses provides a survival advantage postnatally.

Enzymatic activities in the placentas of SeNP-exposed fetuses were compared to the activities observed in the placentas of control fetuses to eliminate the possible effects of vit C, which was present in the solvent for SeNPs. In line with the antioxidant action of vit C [49], lower SOD and GST activities were observed in control placentas in relation to intact controls. To ensure that all animals were subjected to the same experimental conditions, the control and SeNP groups received daily 0.125 M vit C (4.4 mg vit C per dose), as used during the synthesis of SeNPs in order to provide a low pH = 3 and to help in SeNP preservation and maintenance of overall stability. Considering that vit C is an antioxidant, we applied it to the control group of animals to compare the SeNP group with the controls. Further, we compared the results observed in SeNP- and NaSe-treated rats with IC and C groups. Vit C is a potent low-molecular mass antioxidant; however, the doses used in rats in some studies significantly differed from 2.5–2.7 mg/kg to 200 mg/kg (considered as a low dose) [50] to 600 mg/kg (considered as a high dose) [51]. We applied 4.4 mg vit C/kg/day to the rats, which is a very small dose, but we cannot exclude the possibility that it influenced the parameters of oxidative stress. Bearing in mind that both the SeNP group and the control group received vit C, we assumed that the observed changes in the oxidative stress parameters in the SeNP group were the result of the influence of Se nanoparticles. At the same time, the differences between SeNP and IC groups could have been influenced by vit C; based on the obtained results, this is a potential direction in our future work. Treatment with BSA-stabilized SeNPs in both male and female placentas caused a significant increase in SOD and GSH-Px activities with respect to the control group. Increased GSH-Px activity, which is a Se-dependent enzyme, is most probably partly a consequence of increased Se concentration as well as an attempt at neutralization of organic hydroperoxides induced by excessive increase in Se concentration from SeNPs [52]. These data indicate that ROS production in the placenta was increased, elevating the activity of the antioxidative defense system to restore the disturbed redox balance. These results are in accordance with the findings of other authors [16] who reported that Se supplementation increased the activity of GSH-Px and the concentration of GSH. The placenta appears to be particularly vulnerable to oxidative stress due to its extensive cell division and high metabolic activity [52]. Placental oxidative stress plays a key role in the pathophysiology of placenta-related disorders, increased fetal resorption rate and smaller fetuses [53]. It is known that with high intakes, both prenatally and during early childhood, Se can function as a prooxidant with negative effects on neurodevelopment [54,55].

In contrast to treatment with SeNPs, NaSe application resulted in increased concentration of GSH in the placenta of female fetuses when compared to intact controls, which points to the placenta’s potential to maintain oxidoreductive equilibrium. These results confirm that Se at an optimal dose plays a critical role in improved antioxidant generation and efficient ROS elimination [56]. The ameliorative effects of increased Se concentration in the placenta, as our results show, could be mediated by improved mitochondrial function and their biogenesis in trophoblastic cells [56]. Our previous finding in rats [57,58] demonstrated the beneficial effects of NaSe on the antioxidative status in some tissues of rats treated with toxic trace metals, primarily Cd. Se has a similar chemical property in cells comparable to sulfur. This is of immense importance for low-molecular-weight S-containing antioxidants such as GSH and SH. However, Se has some advantages over S. The key differences are in the way the element is incorporated into active sites of enzymes. S as thiol is incorporated into proteins via cysteine (Cys-SH), while Se as a selenol is incorporated into proteins as selenocysteine (Cys-SeH). This difference results in a much greater nucleophilicity of Se-cysteine, which makes it much more reactive to hydroperoxides than Cys-SH [38]. PCA additionally supports the fact that Se preparations lead to a change in the placental redox balance. Considering all analyzed components of the antioxidant defense system, our results show that PC1 discriminated the SeNP and NaSe groups from the IC and C groups of animals; at the same time, PC2 discriminated the NaSe group from all other animal groups. These results indicate that there is a clear difference in placental metabolism between intact control and control animals. Both groups of control animals showed similar trends of changes in oxidative stress parameters. On the other hand, animals treated with BSA-stabilized SeNPs exhibited a different pattern of placental response compared to animals treated with highly reactive NaSe.

Selenium significantly accumulated in the placenta of SeNP and NaSe groups of animals. Even in the control group a significant accumulation of Se was detected when compared to the intact control. The accumulation of Se was about two-fold greater in the placenta of SeNP animals than in animals treated with NaSe. According to the manufacturer’s declaration, standard chow pellets which were fed to the rats contain 0.3 mg Se/kg. As the average daily food intake of rats was about 15 g, during 20 days of experiments, the rats consumed about 300 g of food and ingested about 0.09 mg of Se daily (about 0.03 mg Se/kg bw/day). Bearing in mind that the rats from all four groups ingested the same amount of Se in food, we conclude that the accumulated Se in the placenta was the outcome of the treatment with Se preparations. However, it is unclear why Se accumulated significantly in the placenta of control rats when compared to the intact control. Previous studies have shown that Se accumulates in all organs, but predominantly in metabolically active tissues such as the liver and kidneys [56,57]. The results of the study of Loeschner et al. [34] showed that both forms of Se were proportionately absorbed, distributed, metabolized and excreted. It is unclear why the BSA-stabilized nano form of Se was two-fold higher in the placenta than the NaSe form. This observation should be investigated in the future; at this point we can only assume that the reason is the redox state of elemental Se in the BSA-stabilized nanoparticles.

## 4. Materials and Methods

### 4.1. Animals and Groups in the Experiment

Experiments were performed according to OECD Guidelines for testing chemicals for a prenatal developmental toxicity study (Test No. 414 Adopted, 14 June 2018) [59] on *Wistar* strain adult female rats (*Rattus norvegicus*) weighing about 220–250 g that were bred at the Institute for Biological Research “Siniša Stanković”–National Institute of the Republic of Serbia, University of Belgrade, Belgrade, Serbia. The animals were maintained under controlled conditions (a 12 h light-dark cycle at 22 °C) with ad libitum access to food (standard rat chow) and tap water. Standard rat diet contained mineral components in the amounts presented in Table 6. Complete chemical composition of rat diet is presented in Appendix A.

Two nulliparous females in estrus were mated with one fertile male. The day when sperm was detected in the vaginal smear was designated as day zero of gestation. The gravid females were randomized into four groups as presented in Figure 7.

Regardless of the fact that BSA stabilizes the nano particle and is chemically unreactive and that vit C reduces Se^+4^ in Se^0^, the control group was treated with equivalent concentrations of BSA and vitamin C as in the SeNP group to avoid their possible effects on the investigated parameters. The treatment of pregnant females started on gestational day 1 and lasted until day 20. On the 21st day of gestation, the fetuses were removed from the uterus and were referred to as 21-day-old fetuses. During work with animals, special attention was paid to minimizing their suffering. Fetal and placental weight were measured, and the sex of each fetus was determined by identification of the ovaries or testicles in the abdominal cavity, a procedure performed to collect fetal material necessary for further analysis. Then the viscera index of the placenta was calculated as follows:Viscera index = Viscera weight (g)/Body weight (g) × 100

All animal procedures complied with the European Communities Council Directive (86/609/EEC) and were approved by the Ethical Committee for the Use of Laboratory Animals of the Institute for Biological Research, Belgrade, Serbia (No. 2-12/13).

### 4.2. Synthesis and Physicochemical Characterization of SeNPs 

The SeNPs were synthesized in the form of a colloidal solution, with a concentration of 640 μg/mL. The synthesis procedure uses vit C in excess, which results in a low pH of the obtained colloidal solution (pH = 3). SeNPs were not separate from this solution because we observed that an excess amount of vit C helped SeNP preservation and improved stability. Thus, we considered that the vit C solution with the same concentration should be used as a control to minimize pH and other biochemical effects during supplementation. The main idea of this study was to compare the effects of Se supplementation from two sources. One is ionic selenite (which is highly reactive) and the other is the nanoform. The general issue in the production of material in nanoform is to prevent agglomeration/aggregation. The most used approach against these unwanted phenomena is the utilization of stabilizing agents. In the Se nanoform, BSA has been shown to be very effective [59]. Another approach in the stabilization with BSA is that the interaction of nanoparticles with biological entities is enhanced. Pure Se can be obtained in nanoform under conditions using laser ablation; however, the stability of this form is questionable. Based on the results obtained in this research, this could be a direction for our future work.

Selenium nanoparticles were synthesized by simple chemical reduction of Na₂SeO₃ (sodium selenite) using ascorbic acid as a reducing agent [60] (Appendix A). To preserve the agglomeration of neutral Se atoms within the nanoscale, bovine serum albumin (BSA) was applied. A 20 mM solution of sodium selenite (12.5 mL) was added dropwise into the reaction vessel containing a mixture of a 0.125 M solution of ascorbic acid (10 mL) and a 8.6% solution of BSA (*w*/*v*, 5 mL). The reduction of Se^4+^ to Se^0^ was confirmed through the color change from transparent to brick-red (Appendix A). Since Se is photosensitive, the vessel was covered with aluminum foil. After 30 min of homogenization on a magnetic stirrer (1000 rpm), the colloidal solution of SeNPs was filtered through a 0.24 μm syringe filter (Millipore, Burlington, MA, USA) and stored in a refrigerator at 4–8 °C. For qualitative characterization of SeNPs, the colloidal solution was lyophilized for about 12 h using the Alpha 1-4 LD plus freeze dryer (Martin Christ, Osterode am Harz, Germany).

#### 4.2.1. Fourier-Transform Infrared (FTIR) Spectroscopy

The FTIR spectrum of lyophilized SeNPs was recorded on the Nicolet iS10 Thermo Scientific spectrometer using the attenuated total reflectance (ATR) mode. The spectrum was obtained within a range of 400–4000 cm^−1^ with a resolution of 4 cm^−1^. Based on the vibrational spectroscopy of lyophilized SeNPs, the presence of several functional groups was confirmed (Appendix A). The most dominant absorption of incident IR radiation occurs at 3280 cm^−1^ in the form of a broad shoulder that combines OH and NH vibrations. In a lower frequency region, the most intense peaks correspond to amide bands (amide I at 1590 cm^−1^, amide III multiple peaks between 1400–1300), and a C-O band at 1035 cm^−1^. All peaks confirmed the presence of BSA as a stabilizing agent.

#### 4.2.2. X-ray Diffraction (XRD) Measurements

The X-ray diffraction measurement was performed on a Philips PW 1050 diffractometer with Cu-Kα radiation (Ni filter). The sample was scanned in the 2θ range of 10° to 80°, with a scanning step width of 0.05° and a scanning speed of 2 s per step. On the XRD diagram recorded from lyophilized SeNP powder (Appendix A), no diffraction peaks were observed, indicating that SeNPs were obtained in an amorphous form.

#### 4.2.3. Morphological Studies 

The morphology of SeNPs was analyzed by a field emission scanning electron microscope (FE-SEM) (Supra 35 VP, Carl Zeiss, Jena, Germany) and a transmission electron microscope (TEM) (2100 microscope, Jeol Ltd., Tokyo, Japan). For FE-SEM analysis, the lyophilized sample was coated with carbon, while for TEM analysis a drop of colloidal solution of SeNPs was placed onto a lacey carbon film supported by a 300-mesh-copper grid and dried at ambient conditions. Appendix A shows an FE-SEM image of SeNPs and Appendix A shows a TEM image of these nanoparticles. As can be seen from these images, SeNPs possess spherical morphology with a diameter of particles of about 50 nm.

#### 4.2.4. Inductively Coupled Plasma Optical Emission Spectrometry (ICP-OES)

The concentration of SeNPs in the obtained colloidal solution was determined as previously described [61] by inductively coupled plasma optical emission spectroscopy (ICP-OES, iCap 6500 Duo instrument, Thermo Scientific, Oxford, UK). A working solution of SeNPs was obtained by dilution with 2.5% HNO3, and the intensity of emission was measured at λ = 196.26 nm. The calibration curve was obtained with a multi-element standard solution, MES-21-1 (AccuStandrad, New Haven, CT, USA) at the following concentrations: 10 ppb, 20 ppb, 50 ppb, 100 ppb, 0.2 ppm, 0.5 ppm, 1 ppm and 2 ppm. Based on the ICP-OES measurement, the concentration of SeNPs in the colloidal solution was 640 μg/mL.

### 4.3. Histological and Stereological Analysis of the Placenta

Image acquisition and stereological measurement were performed using a microscope (Olympus, BX-51, Olympus, Tokyo, Japan) equipped with a microcator (Heidenhain MT1201, Heidenhain, USA) to control movements in the z-direction (at 0.2 µm accuracy), a motorized stage (Prior, Prior Scientific Inc., Rockland, MA, USA) for stepwise displacement in the x-y direction (at 1 µm accuracy), and a CCD video camera (PixeLink, Pix eLINK, Gloucester, ON, Canada) connected to a 19” computer monitor. Image acquisition and stage movement were controlled by the newCAST stereological software package (Visiopharm Integrator System (VIS), ver. 3.2.7.0; Visiopharm, Hørsholm, Denmark) running on a personal computer. Placentas from each pregnant female scheduled for histological examination were quickly isolated considering the sex of the fetus (on average, two or three placentas from male and female fetuses from each pregnant female were obtained). The placentas of the remaining fetuses were distributed to determine the tissue Se level and the parameters of oxidative stress (separated by sex also). The determination of the sex of the fetuses was based on the presence of the testes or ovary after opening the abdominal cavity for collection of fetal tissues during dissection. The placentas were fixed with Bouin’s solution for 48 h. After embedding in Histowax (Histolab Product AB, Göteborg, Sweden), each tissue block was serially sectioned at a 12 μm thickness on a rotary microtome (RM 2125RT Leica Microsystems, Wetzlar, Germany) and stained with hematoxylin and eosin. The placental volume was estimated using Cavalieri’s principle, i.e., systematic sampling of every 40th section with random position for the first section [62]. After checking the section thickness by applying the Block Advance (*BA*) method, the area of the selected sections was determined [63,64]. The total volume of each placenta was calculated as follows:$$\overset{⌢}{V}pl=a(p)\cdot BA\cdot \sum_{i=1}^{n}Pi$$
where *a*(*p*) is the area associated with each sampling point (1,242,837.46 µm^2^), *BA* is the mean distance between two consecutively studied sections (480 µm), *n* is the number of sections studied for each placenta and *SPi* is the sum of points hitting a given target.

To express the percentage (percentage share) of layers in the placenta on the 21st day of gestation, four central sections were analyzed per placenta, with a spacing of 120 µm between sections. This quantification provided insight into possible structural changes in the placentas after treatments. The morphometric assessment was performed using 4× magnification of the objective. The counting area was defined using a mask tool. An interactive test grid with uniformly spaced test points for histomorphometric assessment was provided by CAST software. Test points hitting the placental labyrinth, basal zone and decidua were determined. Volume densities (VV) were calculated as the ratio of the number of points hitting each tissue component divided by the number of points hitting the reference space, i.e., the analyzed section: VV (%) = Pp/Pt × 100
where Pp is the counted number of points hitting the tissue component and Pt is total number of points of the test system hitting the reference space.

In pregnant females in which miscarriage was detected on the 21st day of gestation, the uteri with/without aborted fetuses were also taken for histological analysis. After fixation in Bouin’s solution for 48 h, the tissues were routinely processed to paraffin blocks. Serial 5 µm-thick sections were stained with hematoxylin and eosin and histologically examined.

### 4.4. Determination of Oxidative Stress Biomarkers in the Placenta

#### 4.4.1. Preparation of Placental Tissues

Placental tissue samples were washed in 0.9% NaCl and immediately frozen in liquid nitrogen (−196 °C) and stored at −80 °C until analysis for no longer than one month. The tissues were minced and homogenized on ice in 5 volumes of 25 mmol/L sucrose containing 10 mmol/LTris-HCl, pH 7.5, supplemented with 1×phosphatase-inhibitor Mix I and 1×protease-inhibitor Mix G at 4 °C [65] using an IKA-Werk Ultra-Turrax homogenizer (Janke and Kunkel, Staufen, Germany) [66]. The homogenates were sonicated for 30 s at 10 kHz on ice to release enzymes [67], followed by centrifugation in a Beckman ultracentrifuge at 100,000× *g* for 90 min at 4 °C [68,69]. Each sample was measured in triplicate and the mean values of three measurements were used for further calculation.

#### 4.4.2. Antioxidant Enzyme Assays 

The activity of SOD was assayed by the epinephrine method [70]. One unit of SOD activity was defined as the amount of protein causing 50% inhibition of the autoxidation of adrenaline at 26 °C and was expressed as specific activity (U/mg protein). CAT activity was evaluated by the rate of hydrogen peroxide (H2O2) decomposition [71] and expressed as µmol H2O2/min/mg protein. The activity of GSH-Px was determined by following the oxidation of nicotinamide adenine dinucleotide phosphate (NADPH), which served as a substrate, with t-butyl hydroperoxide [72], and expressed in nmol NADPH/min/mg protein. The activity of GST toward 1-chloro-2,4-dinitrobenzene (CDNB) was determined by the method of [73] and expressed as nmol GSH/min/mg protein. The method is based on the reaction of CDNB with the SH group of GSH, which is catalyzed by GST contained in the samples. The activity of GR was measured using the method of [74], based on the capability of GR to catalyze the reduction of oxidized glutathione (GSSG) to reduced glutathione (GSH) using NADPH as a substrate in phosphate buffer (pH 7.4). GR activity was expressed as nmol NADPH/min/mg protein. Protein concentrations in the supernatants were determined according to the method of [75] using bovine serum albumin as a standard and expressed in mg/g wet mass. The activities of the antioxidant enzymes were measured simultaneously in triplicate for each sample using a Shimadzu UV-1800 spectrophotometer with a temperature-controlled cuvette holder at 37 °C. All chemicals were obtained from Sigma-Aldrich (St Louis, MO, USA).

#### 4.4.3. Determination of Nonenzymatic Antioxidants 

The concentration of total GSH was determined as described [76] and expressed as nmol/g of tissue. The concentrations of SH groups were determined using DTNB according to the described method [77] and expressed in µmol/g wet mass. 

#### 4.4.4. Analysis of Se in the Placenta

Analysis of Se content in the placenta was performed as described previously [78]. Each placental sample was precisely weighed on an analytical balance and transferred to a microwave cuvette. Four milliliters of nitric acid (65%) and 1 mL of hydrogen peroxide (30%) were added to each microwave cuvette. Microwave digestion was performed at 180 °C (the temperature was gradually increased to 180 °C for 20 min and remained at the same value for an additional 10 min). Next, the cooled, digested samples were diluted with Milli-Q water into 25 mL normal vessels. Se was quantified using inductively coupled plasma mass spectrometry (ICP-MS, iCAP Qc, Thermo Scientific, UK) in the optimized mode of action and by applying internal standardization. Good linearity (R > 0.999) was obtained in the range from 1 to 300 µg/L. The accuracy of ICP-MS was controlled by SRM (Seronorm™ Trace Elements Whole Blood Level-1, Sero, Billingstad, Norway) and the recorded recovery values were in the range of 96.6 to 105%.

#### 4.4.5. Statistical Analysis

The data were presented as mean values ± standard deviations (SDs). The normality of the data was checked by the Kolmogorov–Smirnov test. One-way analysis of variance (ANOVA) was performed to determine all interactive effects between the investigated tissues. When an interactive effect was observed, Fisher’s least significant difference (LSD) post hoc test was used to obtain significant differences among the means for equal N (sample number). Principal component analysis (PCA) was implemented to statistically determine the differences between the investigated groups based on all investigated antioxidant defense biomarkers and to examine variables that significantly contributed to differences in the investigated parameters. A minimum significance level of *p* < 0.05 was accepted for all cases. All data were processed using the statistical package Statistica 10.0.

## 5. Conclusions

The central idea of this study was to compare the effects of Se supplementation from two sources, one a nanoform and the other ionic selenite (which is highly reactive). The objective was to compare the effects of BSA-stabilized Se nanoparticles (SeNPs) and inorganic sodium selenite (NaSe) supplementation on the histological structure of the placenta, oxidative stress parameters and the total placental Se concentration in *Wistar* rats during pregnancy. SeNPs and NaSe, when administered daily by gastric gavage to the pregnant females in the same dosage of 0.5 mg/kg bw/day, influenced gestation and oxidative stress parameters in the placenta. No sex differences between fetuses were detected in the investigated parameters after treatment with both SeNPs and NaSe. In contrast to NaSe, SeNPs could cause embryo-fetal lethality. The accumulation of Se in the placenta was highest in SeNP-treated animals, which could be the cause of its toxicity. Our findings show that the mode of action of equivalent concentrations of Se in BSA-stabilized SeNPs and highly reactive NaSe depended on its redox state and chemical structure. All obtained data indicate that the increased bioavailability of Se originated from SeNPs (Se^0^) rather than NaSe (Se^0^), which exerts a harmful effect on intrauterine development.

## Figures and Tables

**Figure 1 ijms-23-13068-f001:**
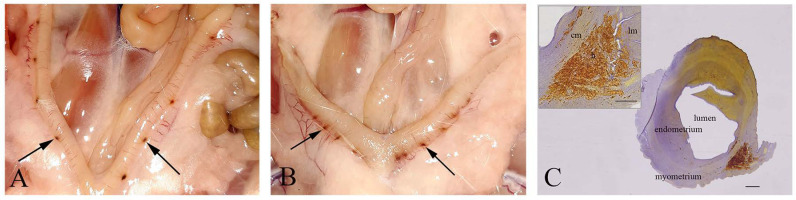
Embryonal resorption. (**A**,**B**)—Pinpoint hemorrhagic placentation sites (arrows) on the uterine wall. (**C**)—Cross-section of the hemorrhagic placentation sites (h); lm—longitudinal muscle cells; cm—circular muscle cells; bar = 200 µm; insert bar = 100 µm.

**Figure 2 ijms-23-13068-f002:**
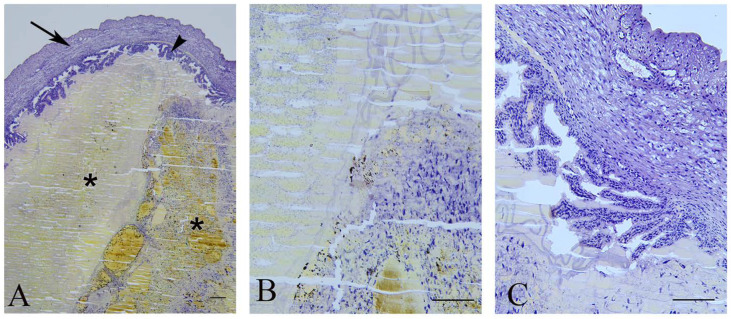
Late miscarriage. (**A**)—Uterus (arrow) with undeveloped fetuses (asterisks) and chorion villi (arrowhead) border arrows); bar = 200 µm. (**B**)—Detail of the remnants of the fetus; bar = 100 µm. (**C**)—Branched chorionic villi; bar = 100 µm.

**Figure 3 ijms-23-13068-f003:**
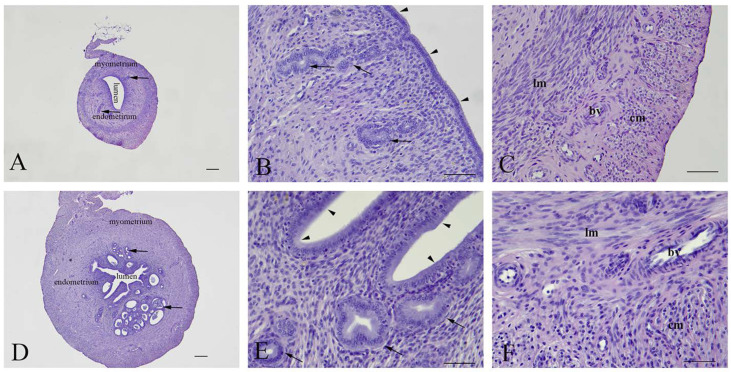
Representative micrographs of uterine cross sections. (**A**–**C**)—Early miscarriage; (**D**–**F**)—late miscarriage. Arrowheads—luminal epithelium; Arrows—endometrial glands; lm—longitudinal and cm—circular muscle layers; bv—blood vessels. (**A**,**D**)—bar = 200 µm; (**B**,**C**,**E**,**F**)—bar = 50 µm.

**Figure 4 ijms-23-13068-f004:**
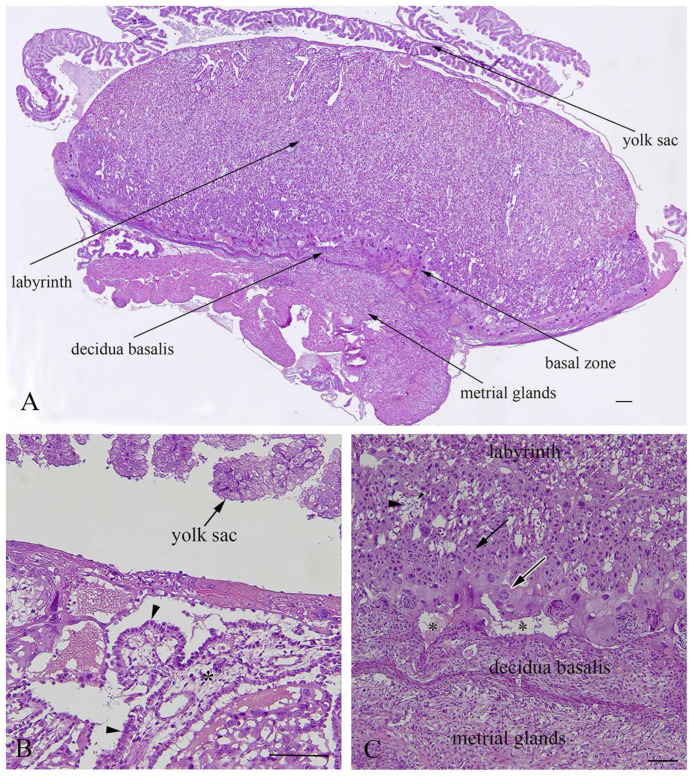
Histological structure of rat placenta at day 21 of pregnancy. (**A**)—Structural layers of rat placenta; (**B**)—large trophoblastic septa with syncytiotrophoblasts and a cytotrophoblast (arrowhead) facing towards maternal blood; (**C**)—basal layer with spongiotrophoblasts (arrow),trophoblastic giant cells (white border arrow), a degenerated glycogen cell (arrowhead), and a regressed decidua basalis with dilated blood vessels (asterisks); (**A**)—bar = 200µm; (**B**)—bar = 100µm; (**C**)—bar = 200µm.

**Figure 5 ijms-23-13068-f005:**
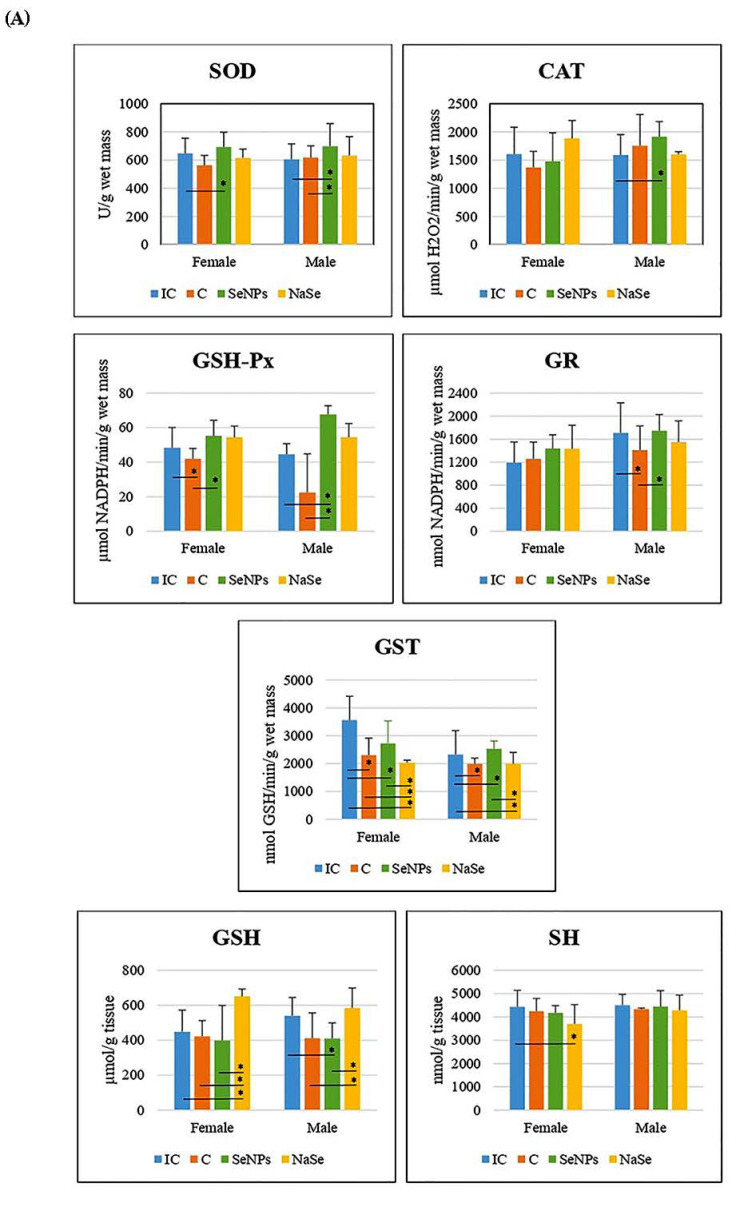
The activities of SOD (U/g wet mass), CAT (U/g wet mass), GSH-Px (U/g wet mass), GR (U/min/g wet mass) and GST (U/g wet mass), and the concentrations of total GSH (µmol/ g tissue) and SH (nmol/g tissue) in the placenta. (**A**)—Differences between treated groups. (**B**)—Differences between fetuses of a different gender. Intact controls-IC; controls-C; Se nanoparticles-treated animals-SeNP; sodium selenite-treated animals-NaSe. The data are expressed as the mean ± S.D. One-way ANOVA post hoc Fisher’s LSD test was used. * Significantly different when *p* < 0.05.

**Figure 6 ijms-23-13068-f006:**
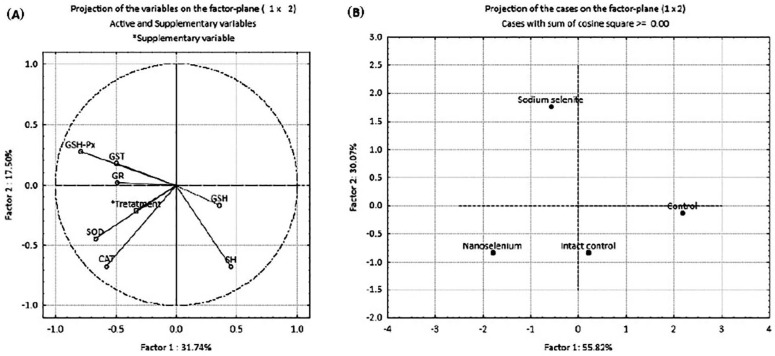
PCA of oxidative stress biomarkers in the placenta of pregnant *Wistar* albino rats in intact controls, controls, nanoselenium-treated animals and sodium selenite-treated animals. (**A**) —projection of the relative contribution of every antioxidant component; (**B**) —projection of groups based on antioxidant defense parameters.

**Figure 7 ijms-23-13068-f007:**
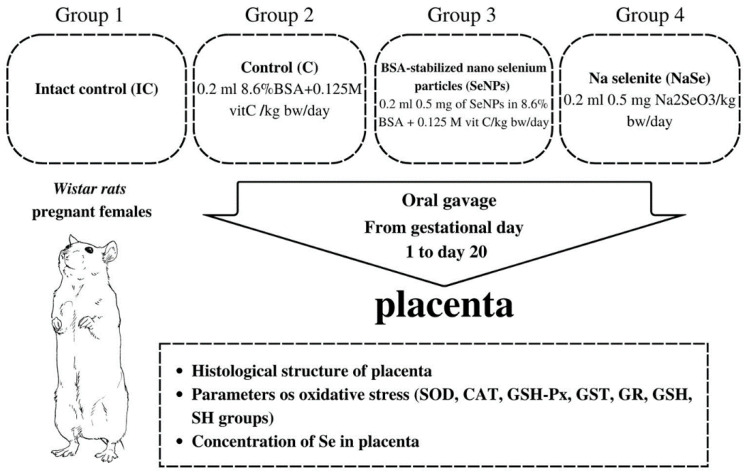
Schematic of the experimental design.

**Table 1 ijms-23-13068-t001:** Fetal weight (g), placental weight (g) and viscera index of the placenta (g/g) after the treatment of pregnant females with SeNPs and sodium selenite. ICF—intact females; ICM—intact males; CF—control females; CM—control males; SeNPF—nano Se females; SeNPM—nano Se males; NaSeF—sodium selenite females; NaSeM—sodium selenite males. Significant differences between groups: ^a^ in respect to ICF; ^b^ in respect to CF; ^c^ in respect to ICM; ^d^ in respect to CM. *p* ˂ 0.05 was considered as significant.

	N^o^ of Fetuses/Dam	Fetal Weight (g)	Placental Weight (g)	Viscera Index of the Placenta (g/g)
IF	13 ± 2	5.08 ± 0.29	0.52 ± 0.04	10.05 ± 0.76
IM		5.48 ± 0.32	0.55 ± 0.05	10.09 ± 1.00
CF	11 ± 2	5.18 ± 0.25	0.49 ± 0.04	9.24 ± 1.05
CM		5.28 ± 0.42	0.51 ± 0.04	9.44 ± 1.41
SeNPF	12 ± 1	4.44 ± 0.33 ^a,b^	0.43 ± 0.06	9.33 ± 1.67
SeNPM		4.67 ± 0.23 ^c,d^	0.48 ± 0.06	9.33 ± 1.33
NaSeF	11 ± 2	6.02 ± 0.17 ^a,b^	0.50 ± 0.06	8.27 ± 1.02
NaSeM		6.26 ± 0.38 ^c,d^	0.55 ± 0.07	8.71 ± 1.19

**Table 2 ijms-23-13068-t002:** The activities of SOD (U/g wet mass), CAT (U/g wet mass), GSH-Px (U/g wet mass), GR (U/g wet mass), GST (U/g wet mass), and the concentrations of GSH (µmol/g tissue) and SH (nmol/g tissue) in the placentas of male and female fetuses: intact controls (IC), controls ©, Se nanoparticles-treated animals (SeNPs) and sodium selenite-treated animals (NaSe). The data are expressed as the mean ± S.D. One-way ANOVA post hoc Fisher’s LSD test was performed with *p* < 0.05 as the level of significance. Significant differences between groups: ^A^ C vs. IC; ^B^ SeNPs vs. IC; ^C^ SeNPs vs. C; ^D^ NaSe vs. IC; ^E^ NaSe vs. C; ^F^ NaSe vs. SeNPs. N—total number of individuals.

N = 40	Gender	Intact Control (IC)	Control (C)	Nanoselenium (SeNP)	Na-Selenite (NaSe)
SOD	Males + Females	634.93 ± 116.68	582.68 ± 76.64 ^A^	695.54 ± 119.17 ^C^	625.36 ± 100.68
CAT	Males + Females	1597.331 ± 395.84	1504.63 ± 422.12	1637.49 ± 472.12 ^B^	224.58 ± 100.44 ^D,E,F^
GSH-Px	Males + Females	46.31 ± 9.02	48.49 ± 15.96	59.75 ± 9.83 ^B^	54.40 ± 6.55
GR	Males + Females	1471.94 ± 513.01	1314.90 ± 330.54	1549.55 ± 287.21	1501.93 ± 337.85
GST	Males + Females	2888.64 ± 1036.71	2191.48 ± 510.01 ^A^	2655.68 ± 649.79 ^C^	2007.50 ± 324.59
GSH	Males + Females	498.70 ± 117.70	419.86 ± 98.19	402.85 ± 162.41 ^B^	610.98 ± 91.05 ^D,E,E^
SH	Males + Females	4467.85 ± 560.79	4295.83 ± 503.85	4267.94 ± 468.55 ^B^	4047.82 ± 661.15

**Table 3 ijms-23-13068-t003:** The activities of SOD (U/g wet mass), CAT (μmol H_2_O_2_/min/g wet mass), GSH-Px (μmol NADPH/min/g wet mass), GR (nmol NADPH/min/g wet mass) and GST (nmol GSH/min/g wet mass), as well as total GSH (µmol/ g tissue) and SH groups (nmol/g tissue) concentrations in placenta. The effects between fetuses gender, as well as between placenta of pregnant Wistar albino rats.

Treatment	Intact Control (IC)		Control (C)		Nanoselenium (SeNPs)		Na-selenite (NaSe)	
Gender	Female	Male	Female	Male	Female	Male	Female	male
SOD	647.56 ± 107.08	605.11 ± 109.90	562.91 ± 70.73	617.27 ± 84.06	693.39 ± 104.15 ^CF^	699.29 ± 160.06 ^BM,CM^	615.00 ± 63.64	632.27 ± 134.41
CAT	1606.79 ± 476.05	1589.45 ± 363.20	1367.10 ± 287.63	1745.30 ± 554.20	1479.16 ± 504.92	1914.56 ± 268.83 ^BM^	1884.63 ± 318.20	1601.83 ± 47.61
GSH-Px	48.41 ± 11.69	44.56 ± 6.2	41.98 ± 6.0	59.90 ± 12.43 ^AM,C^*	55.21 ± 9.08 ^CF^	67.71 ± 4.99 ^BM,CM^	54.40 ± 6.55	54.40 ± 8.00
GR	1189.34 ± 361.94	1707.39 ± 524.58 ^I^*	1258.84 ± 290.95	1410.77 ± 419.78 ^AM^	1436.61 ± 239.18	1747.17 ± 281.41 ^CM^	1432.47 ± 409.26	1548.23 ± 369.47
GST	3562.50 ± 852.25	2327.08 ± 857.89	2306.25 ± 608.35 ^AF^	1990.62 ± 205.74 ^AM^	2726.77 ± 805.11 ^BF^	2531.25 ± 280.35 ^CM^	2031.25 ± 91.68 ^DF,EF,GF^	1991.67 ± 408.50 ^DM,GM^
GSH	448.63 ± 123.43	540.43 ± 104.50	422.55 ± 89.88	411.21 ± 144.99	398.96 ± 199.77	409.66 ± 89.52 ^BM^	650.42 ± 42.78 ^DF,EF,GF^	584.69 ± 114.34 ^EM,GM^
SH	4428.97 ± 711.69	4500.25 ± 470.09	4243.98 ± 549.35	4328.43 ± 53.52	4166.80 ± 312.12	4440.93 ± 686.54	3695.46 ± 833.54 ^DF^	4282.76 ± 655.54

The data are expressed as the mean ± S.D. The one-way ANOVA post hoc Fisher’s LSD (least significant difference) test was used with *p* < 0.05 as the level of significance. Significant differences between treated groups: ^A^ C vs IC; ^B^ SeNPs vs IC; ^C^ SeNPs vs C; ^D^ NaSe vs IC; ^E^ NaSe vs C; ^G^ NaSe vs SeNPs; Significant differences between gender groups: ^F^ Female; ^M^ Male vs Male; ^I^* Intact female vs Intact male; ^C^*Control female vs Control male.

**Table 4 ijms-23-13068-t004:** Factor coordinates of the variables, based on correlations onto the principal components (PC) in placenta of pregnant *Wistar* albino rats. (Intact controls, controls, gavaged with bovine serum albumin and vitamin C/day, nano Se-treated animals, gavaged with 0.5 mg Se in the form of SeNPs/kg/body weight/day and sodium selenite (Na-selenite)-treated animals, gavaged with 0.5 mg Se in the form of Na_2_SeO_3_/kg/body weight/day.) The parameters and correlations that contributed most to the separation are marked in bold and marked with asterisk (*).

	PC1	PC2	PC3	PC4	PC5	PC6	PC7
**SOD ***	**−0.621456 ***	0.278264	−0.007269	0.044775	−0.044528	−0.488686	−0.230895
**CAT ***	**−0.682013 ***	**−0.666915 ***	−0.167632	0.129144	−0.092093	0.304164	−0.265914
**GSH-Px ***	**−0.798461 ***	**−0.437845 ***	0.188150	−0.047289	0.491792	−0.116416	0.261540
GR	−0.487445	−0.256611	0.670431	−0.003598	−0.531476	0.093257	0.145790
GST	−0.496050	0.183119	−0.720209	0.273873	−0.247137	0.048427	0.251540
GSH	0.352570	−0.163949	0.214057	0.885287	0.031405	−0.135105	−0.065733
**SH**	0.448887	**−0.670984 ***	−0.185965	−0.220622	−0.340460	−0.379719	0.070178
* Treatment	−0.338659	−0.210221	0.364973	0.360188	0.168948	−0.010868	0.140813

**Table 5 ijms-23-13068-t005:** The concentration of Se in the placenta of female and male rat embryos. The groups were as follows: intact controls (IC); controls (C); Se nanoparticles-treated group (SeNPs); sodium selenite-treated group (NaSe). The data are expressed as the mean ± S.D. The independent *t*-test by groups was used for comparisons between a group with *p* < 0.05 as the level of significance. Significant differences between treated groups: IC concerns the intact control; C concerns the control group. * SeNP vs. NaSe. ^IC^: with respect to the intact controls; ^C^: with respect to the controls.

Selenium Concentration	Intact Control (IC)	Control (C)	Se Nanoparticles (SeNP)	Sodium Selenite (NaSe)
Female + Male	149.30 ±12.48	193.50 ± 15.40 ^IC^	722.30 ± 201.55 ^IC,C^	385.31 ± 135.36 ^IC,C,^*
Female	148.93 ± 14.69	191.82 ± 6.25 ^IC^	657.69 ± 187.68 ^IC,C^	371.21 ± 94.85 ^IC,C,^*
Male	149.67 ± 15.85	204.66 ± 18.33 ^IC^	734.82 ± 203.92 ^IC,C^	350.27 ± 94.85 ^IC,C,^*

**Table 6 ijms-23-13068-t006:** Mineral composition of the rat diet used in the experiment.

Mineral	mg/kg
Iron	min 600
Zinc	min 30
Manganese	min 60
Copper	min 15
Iodine	min 0.15
Selenium	min 0.1

## Data Availability

The data presented in this study are available at a reasonable request from the corresponding author.

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
