# Peer review of "The Effects of BSA-Stabilized Selenium Nanoparticles and Sodium Selenite Supplementation on the Structure, Oxidative Stress Parameters and Selenium Redox Biology in Rat Placenta"

_ijms, 2022, doi:10.3390/ijms232113068_

Round 1

Reviewer 1 Report

Please find my comments in the attached file.

Regards

Author Response

Responses to the reviewers’ comments

The reviewers comments were very useful and all have been incorporated in the revised version of our MS. We also greatly appreciate the time the reviewers spent and their desire to improve this manuscript. Language editing was again performed by a native English language speakers Ms. Myra Isabel Poznanović (née Macpherson), and by Dr. Goran Poznanović, Ph.D., an expert in the field of molecular biology. The reviewers’ suggestions are addressed point-by-point. In the Word document, all corrections in the text are marked using the “Track Changes” function.

Answers to the Reviewer #1:

We are grateful to the reviewer for the comments and constructive suggestions.

Reviewer request: the writing style and expression is less engaging and at times very raw. I, therefore, recommend an efficient round of editing.

Answer to the Reviewer: As suggested by Reviewer, we have tried to improve the writing style and expression wherever possible to make the text more understandable for reading.

  1. Reviewer request: Title has to be concise yet informative. In its current form, it is vague. Authors have used two different sources of selenium (org vs. inorg.), therefore, this should be reflected in the title and redundant words/phrase like histological structures should be dispended.

Answer to the Reviewer: As requested by the reviewer, we have now changed the title to make it clearer. We believe it would not be suitable to use the term organic Se for SeNP because in Nano form Se is in an elemental state, without chemical bonding to BSA (confirmed by FTIR). BSA is a stabilizing/coating agent. So, the new title of our work is: “The effects of BSA-stabilized selenium nanoparticles and inorganic sodium selenite supplementation on the structure, oxidative stress parameters and selenium redox biology in rat placenta”. We also removed unnecessary phrases from the title.

Abstract:

  • Reviewer request: I suggest complete overhauling of this section.

Answer to the Reviewer: We rearranged the abstract as much as possible.

  • Reviewer request: I would rather appreciate if authors can introduce selenium first and then go on to focus on its
    relevance to pregnancy. This will help general readership.

Answer to the Reviewer: According to the reviewer's recommendation, we first introduced the general role of selenium, and then its role in pregnancy.

  • Reviewer request: Also, information related to experimental design and concentration of selenium supplementation is missing and must be included in the abstract.

Answer to the Reviewer: There is already in the abstract a brief description of the experimental design, groups that have been formed and doses of organic and inorganic Se that female rats received by oral gavage.

  • Reviewer request: Line 22: “morpho-anatomical characteristics of the mothers and 22 fetuses”. This is vague and should be elaborated for clarity.

Answer to the Reviewer:  Since we only measured maternal and fetal weights, we decided to remove this phrase from the text.

  • Reviewer request: Line 31: “Embryo-fetal lethality and a disturbed redox balance in the placenta were shown using SeNPs (Se0 ).” Again this is confusing.

Answer to the Reviewer: We agree with the reviewer that this part of the text is confusing, and we have tried to be clearer and more precise.

Introduction:

  • Reviewer request: Lines 38-39: please ad suitable reference.

Answer to the Reviewer: We added a suitable reference No. 1.

  • Reviewer request: Line 39: Please rephrase this “As a member of more than twenty- 39 five selenoproteins, a small amount of Se

Answer to the Reviewer: We rephrased this sentence as suggested.

  • Reviewer request: Lines 43-46: Authors highlight the impact of Se deficiency, but fail to stress that the response curve of Se is U shaped and as such its high concentration are reportedly detrimental.

Answer to the Reviewer: As suggested, we mentioned the U-shaped curve of the Se response and cited appropriate reference No. 4.

  • Reviewer request: Line 56-57: “. Considerable effort has been invested to determine a form of Se with high bioavailability and low toxicity”. Please support this with a citation. But generally we all know that organic selenium is more bioavailable. Story of nanoparticles is different than what we argue for organic vs. inorganic forms

Answer to the Reviewer: The sentence is more precisely written and is supported by a citation as suggested.

  • Reviewer request: Lines 73-74: “The placenta has an active role in Se transport during the entire course of intrauterine development”. Please support this statement with a citation. The same for line 80.

Answer to the Reviewer: The statement is supported by citation No. 20. Also, the statement (L. 80) is supported by citation No. 23.

  • Reviewer request: Also line 82-85: “One of the reasons for oxidative stress is the deterioration of syncytiotrophoblasts in the presence of high oxygen levels, resulting in cell vacuolization, a reduction of surface microvilli and a decline in the number of mitochondria without ensuing damage to cytotrophoblasts and stromal cells.” Please add suitable citation.

Answer to the Reviewer: Appropriate citation is added (No. 24) as suggested.

  • Reviewer request: Lines 94-97: Bearing in mind that Se dosage is of crucial importance [24], the assessment of fetotoxicity after nanoparticle application during pregnancy is of imperative, since developing organisms are the most vulnerable and the induced alterations can cause permanent and lifelong health consequences [25,26]. Please rephrase to improve clarity and flow.

Answer to the Reviewer: To improve clarity and flow, we reformulated the sentence as suggested.

  • Reviewer request: Now unfortunately text in lines 101-115 should be dispensed. This is more of methodical information. Authors need to find gap in our understanding if this particular facet and then objectively mention what they are going to do in the current study

Answer to the Reviewer: The text in L. 101-115 is ‘dispensed’. For this section, we have written a clearer text that is in accordance with the changes in the title and abstract, avoiding methodological information.

Results:

  • Reviewer request: Overall, this section needs significant improvement. Most of the sentences are raw and confusing. Almost all important information is buried in long threads of text, which should be optimized. Moreover, histological sections are poor and must be improved. Table titles and footnotes should be improved. Importantly, it is highly recommended that tables that were kept as supplementary files should be added in the main text. This is the essence of your study and must be displayed in the main text.

Answer to the Reviewer:

As suggested by the reviewer, this part is significantly changed. We tried to avoid raw and confusing sentences, we optimized the text and improved the histological section. Table titles and footnotes have been improved, but in such a way that they remain self-explanatory. We added the tables from the supplementary material to the main text as suggested.

Discussion:

  • Reviewer request: Lines 307-317 should be revised. This is simple repetition of the results and should be avoided. To make is biologically relevant discussion, authors need to critically discuss their finding in light of mainstream literature. Focused on condensed information will be very helpful to readership.

Answer to the Reviewer: We avoided repetition of the text from the results section.

Material and Methods:

  • Reviewer request: Why this specific strain of rats was used in this study? It has shown in mice that different strains
    have different reproductive features; some of them are not suitable for reproductive studies.

Answer to the Reviewer: Our study was planned according to the OECD GUIDELINE FOR TESTING OF CHEMICALS for Prenatal developmental toxicity study (Test No. 414 Adopted: 25 June 2018), where it is stated that the preferred rodent species is the rat. The Wistar strain was selected as one of the most used outbred strains in much medical and biological research and extensive physiological, toxicological and reproductive data are available for this strain (Laboratory Animal Medicine). Animals of Wistar strain are smaller (in comparison to another widely used outbred rat strain – Spague Dawley), and hence easier to handle. Having in mind that in our experiment feeding of pregnant females was required, we chose the Wistar strain (Laboratory Animal Medicine).

LABORATORY ANIMAL MEDICINE, 2nd edition; Biology and Diseases of Rats by Dennis F Kohn and Charles B., Clifford Chapter 4; Copyright 2002, Elsevier Science (USA). ISBN 0-12-263951-0.

Numerous studies of developmental toxicity used this strain of rats. Some of them are:

  • Catlin NR, Bowman CJ, Campion SN, Cheung JR, Nowland WS, Sathish JG, Stethem CM, Updyke L, Cappon GD. Reproductive and developmental safety of nirmatrelvir (PF-07321332), an oral SARS-CoV-2 Mpro inhibitor in animal models. Reprod Toxicol. 2022 Mar;108:56-61. doi: 10.1016/j.reprotox.2022.01.006.
  • Harris SB, Hardisty JF, Hayes AW, Weber K. Developmental and reproductive toxicity studies of BIA 10-2474. Regul Toxicol Pharmacol. 2020 Mar;111:104543. doi: 10.1016/j.yrtph.2019.104543.
  • Kumar P, Lomash V, Jatav PC, Kumar A, Pant SC. Prenatal developmental toxicity study of n-heneicosane in Wistar rats. Toxicol Ind Health. 2016 Jan;32(1):118-25. doi: 10.1177/0748233713498438.
  • Pratap Reddy, B. P. Girish, P. Sreenivasula Reddy, Reproductive and paternal mediated developmental toxicity of benzo(a)pyrene in adult male Wistar rats, Toxicology Research, Volume 4, Issue 2, March 2015, Pages 223–232, https://doi.org/10.1039/c4tx00121d

  • Reviewer request: What was logic behind using Vit C? This should be clearly mentioned in the methods and discussion chapters. What was its impact on the results? Why authors did not use pure form of Se?

Answer to the Reviewer: The SeNPs were synthesized in the form of a colloidal solution, with a concentration of 640 μg/mL. The synthesis procedure requires the use of Vit C in excess, which further results in a low pH of the obtained colloidal solution (pH=3). SeNPs were not separate from this solution because we observed that an excessive amount of Vit C helps in SeNP preservation and overall stability. Thus, we considered that the Vit C solution with the same concentration should be used as a control to minimize the effects of pH and other biochemical effects during supplementation. The main idea of this study was to compare the effects of Se supplementation from two sources. One is ionic selenite (highly reactive) and the other is a nanoform. The general issue in the production of material in the nanoform is preventing agglomeration/aggregation. The most used approach to prevent these unwanted phenomena is the utilization of stabilizing agents. In our Se nanoform, BSA has proven to be very effective (https://doi.org/10.3389/fbioe.2020.624621). Another approach in the stabilization with BSA is that it often enhances the interaction of nanoparticles with biological entities. Pure Se can be obtained in nanoform under unusual conditions such as by use of laser ablation. However, the stability of this form will be very questionable. Based on the results obtained in this research, this could be a direction for our future work.

  • Reviewer request: There is possibility that results are skewed due to addition of Vit C? Authors have added some
    justification but that not enough and should be supported by reference.

Answer to the Reviewer: In order that all experimental animals be subjected to the same experimental conditions (except IC group) they received the same substances. Control, SeNP and NaSe groups received 0.125M Vit C (about 4.4 mg Vit C per dose) which was used during the synthesis of SeNPs to provide a low pH=3 and to help in SeNP preservation and to provide overall stability. Considering that it is an antioxidant, we decided to introduce a control group of animals that received BSA and Vit C, and to compare other groups with the control that received Vit C. Further, we compared the results of SeNP- and NaSe-treated rats with those of IC and C groups. Such an experimental design was used to ensure identical experimental conditions for all treated animals. Thereby, we obtained some differences. While it is well known that Vit C is a potent low molecular mass antioxidant, the doses used in some studies in rats significantly differed, from 2.5-2.7 mg/kg (https://doi.org/10.7150/ijms.17681) to 200 mg/kg (considered as a low dose), to 600 mg/kg (considering as a high dose) (https://doi.org/10.1007/s00580-012-1430-9). We applied 4.4 mg Vit C/kg/day to the rats, which is a very small dose, but we cannot exclude the possibility that it influenced the composition of the components of the oxidative stress parameters. Based on the results obtained in this research, this could be another direction for our future work.

  • Reviewer request: It is better that authors add a schematic for experimental design and avoid long text in the main

Answer to the Reviewer: As suggested by Reviewer, instead of textual explanation of experimental procedure, we added appropriate schematic diagram.

  • Reviewer request: Why these concentrations of Se were used?

Answer to the Reviewer: According to OECD Guidelines, the concentration of Se used in our experiments was 0.5 mg/kg bw/day and was selected as it was previously described as being supranutritional and non-lethal in rodents (https://doi.org/10.1016/j.taap.2013.01.028;  https://doi.org/10.1039/c3mt00309dd; https://doi.org/10.1007/s12011-017-0980-8).

  • Reviewer request: What is the composition of diet used for rodents?

Answer to the Reviewer: According to the manufacturer declaration, the composition of the rat diet is as follows:

Manufacturer:

Name of food: Complete mixture for laboratory rats (chow pellets)

Manufacturer: D.O.O. GEBI, ÄŒantavir 24220, Maršala Tita 46, Serbia

Tel.: (+381) 24-782-525, Fax: (+381) 24-782-526. www.gebi.rs; office@gebi.rs

 CHEMICAL COMPOSITION OF FOOD

Proteins    

min   20.00%

Vit B2

min 7.00 mg/kg

Fat            

min     5.00%

Vit B6

min 6 mg/kg

Humidity  

max 11.5%

Vit B12

min 0.05 mg/kg

Cellulose

max 5.00%

Folic acid

min 1.0 mg/kg

Ashes

max 8.00%

Choline chloride

1.000.00 mg/kg

Calcium

min 0.95%

Niacin

20.00 mg/kg

Phosphorus

min 0.70%

Pantothenic acid

min 10.00 mg/kg

Lysine

min 0.60%

Methionin

min 0.60%

MINERALS

Tryptophan

max 0.15%

Zinc

min 30 mg/kg

Copper

min 15.00 mg/kg

VITAMINS

Iron

min 60 mg/kg

Vit A

min 12.000 U/kg

Manganese

min 60 mg/kg

Vit D3

min 2.000 U/kg

Iodine

min 0.15 mg/kg

Vit E

min 30.00 mg/kg

Selenium

min 0.30 mg/kg

Vit B1

min 4.00 mg/kg

Antioxidants: E-321, E-320, E-338, E202

min 125 mg/kg

Quality control is performed by: Institute for Animal Husbandry, Laboratory for Chemistry and Microbiology, Highway No 16 Street, Belgrade, Serbia.

Veterinary control number: RS-30-004

  • Reviewer request: Please add if you performed any analyses (other than the mineral components) of diet used in this study?

Answer to the Reviewer: We did not perform an additional nutritional analysis. Food from the above mentioned certified food manufacturer and the composition and quality of the food are controlled by the certified reference laboratory.

  • Reviewer request: Figure 7. Can be moved to supplementary material.

Answer to the Reviewer: As suggested, Figure 7 was moved to the Supplementary material and now it is Figure S3 (Supplementary materials).

  • Reviewer request: Add citation for lines 517-521.

Answer to the Reviewer: Citation was added as suggested (Ref 59).

Conclusions:

  • Reviewer request: This sections needs to be rewritten and only conclusion based on this finding should be focused. It is unclear what the central objective was and whether it was achieved or not. Lines 616-17 should be toned down.

Answer to the Reviewer: As suggested by both reviewers, we reformulated the Conclusions section and what is the central objective, and have highlighted which compounds were used, which substance had more pronounced effects at the same concentrations, which accumulated to a higher extent, and which is more bioavailable. We avoided merely documenting the effects of the applied substances.

Once again, thank you for your time and constructive suggestions to improve quality of our manuscript.

Sincerely,

Slađan Pavlović, Ph.D.

Principal Research Fellow

Reviewer 2 Report

The manuscript entitled ,,The effects of selenium nanoparticles and sodium selenite sup- 2 plementation on the histological structure, oxidative stress pa- 3 rameters and selenium redox biology in rat placenta" is well written and bring the important role of microelements such as selenium in pregnancy.

The following observations are necessary:

Abstract - replace "gravid female" with "pregnant female"

Introduction  - line 38 - correct "which is" with "which are" 316 (and other places)

Line 316 - please replace the word "dams" ; do you mean animals?

line 419 - please offer the Ethical Committee Approvement for your experimental study

Conclusions 

Please reformulate the conclusions more clear. The readers should clear understand, which compound can be used, which is more efficient what is the administration route. The clear differences between your study molecules should be emphasized. Is improper to mention in your conclusions that you only documented their effects. 

Author Response

Responses to the reviewers’ comments

The reviewers comments were very useful and all have been incorporated in the revised version of our MS. We also greatly appreciate the time the reviewers spent and their desire to improve this manuscript. Language editing was again performed by a native English language speakers Ms. Myra Isabel Poznanović (née Macpherson), and by Dr. Goran Poznanović, Ph.D., an expert in the field of molecular biology. The reviewers’ suggestions are addressed point-by-point. In the Word document, all corrections in the text are marked using the “Track Changes” function.

Answers to the Reviewer 2:

            First of all, I wish to thank to reviewer 2 for the effort he made and for the time the reviewer spent to improve our manuscript.

  1. Reviewer request: Abstract - replace "gravid female" with "pregnant female".

Answer to the Reviewer: We replaced “gravid female” with “pregnant female” as suggested.

  1. Reviewer request: Introduction  - line 38 - correct "which is" with "which are" 316 (and other places)

Answer to the Reviewer: We corrected “which is” with “which are” in L. 38 as suggested, but at the suggestion of the English proofreader, no correction was needed elsewhere.

  1. Reviewer request: Line 316 - please replace the word "dams"; do you mean animals?

Answer to the Reviewer: Throughout the text, we replaced the term “dams” with the term “pregnant females” as suggested.

  1. Reviewer request: line 419 - please offer the Ethical Committee Approvement for your experimental study.

Answer to the Reviewer: In the attachment, we submit the decision of the Ethics Committee of the Institute for Biological Research “Siniša Stanković” – National Institute of the Republic of Serbia, Decree No. 01-03/19 that was obtained specially for this study from the Ethics Committee of the Institute for Biological Research on March 11, 2019 (No 01-615), (in Serbian). If the reviewer requests, we will submit the decision of the ethics committee after translation into English.

  1. Reviewer request: Conlusions: Please reformulate the conclusions more clear. The readers should clear understand, which compound can be used, which is more efficient what is the administration route. The clear differences between your study molecules should be emphasized. Is improper to mention in your conclusions that you only documented their effects.

Answer to the Reviewer: As suggested by both reviewers, we reformulated the Conclusions section and highlighted which compounds were used, which substance has more pronounced effects at the same concentrations, which accumulates more, and which is more bioavailable. We avoided merely documenting the effects of the applied substances.

Once again, thank you for your time and constructive suggestions to improve quality of our manuscript.

Sincerely,

Slađan Pavlović, Ph.D.

Principal Research Fellow

Round 2

Reviewer 1 Report

Most of my comments have been addressed, however, there are still a few comments which I hope will be incorporated by the authors. 

Please remove "inorganic" from title as "sodium selenite" is itself an inorganic form of Se. 

Wherever, Se is used at the beginning of a sentence, it should be in full, and abbreviated (Se) at the rest. Please remove word "diagram from legend of figure 7. Keep only schematic, as both have the same meaning.

Line 581 and elsewhere: Please substitute the word “experimental”, because you have control group as well. Simply write them groups.  Check this throughout the text.

Please provide the diet composition in the supplementary material. This information, which otherwise ignored by many of us, is very essential for replication and reproducibility of results.

Rat “food” should be rat “diet”. Please check.

I disagree to the contention of authors regarding dosage of selenium. There have been varying opinions regarding what is actual supra-nutritional levels. Prof. Margaret Rayman has rightly said “Selenium will surely keep us guessing for some time to come”.

Although there has been some advancement in recent years, therefore, previous recommendations (from older studies) using rodent models should be very carefully evaluated. It has been reported recently that different levels of Se have different effects on expression of selenoproteins and that also depend on organ/context in which they are being studied.

For ensuring the consistency of nomenclature of selenoproteins, authors are suggested to review article of Dr. Vadim et al. 10.1074/jbc.M116.756155. We all have been strong advocate of bringing consistency in nomenclature.

Please remove “native English language speaker” from acknowledgment, as their names are enough. We better be inclusive.

Best regards

Author Response

Responses to the reviewer comments

We are grateful to the reviewer for the comments and constructive suggestions. All reviewer comments have been incorporated in the revised version of our MS. We again greatly appreciate the time the reviewers spent and their desire to improve this manuscript. Language editing was again performed. The reviewers’ suggestions are addressed point-by-point. In the Word document, all corrections in the text are marked using the “Track Changes” function.

Answers to the Reviewer #1:

  • Reviewer request: Please remove "inorganic" from title as "sodium selenite" is itself an inorganic form of Se. 

Answer to the Reviewer: “Inorganic” was removed from the title.

  • Reviewer request: Wherever, Se is used at the beginning of a sentence, it should be in full, and abbreviated (Se) at the rest. Please remove word "diagram from legend of figure 7. Keep only schematic, as both have the same meaning.

Answer to the Reviewer: We have carefully reviewed the text and wherever Se is at the beginning of a sentence we have used the full name, and abbreviated  (Se) at the rest. The word “diagram” was removed from legend of Figure 7.

  • Reviewer request: Line 581and elsewhere: Please substitute the word “experimental”, because you have control group as well. Simply write them groups. Check this throughout the text.

Answer to the Reviewer: In Line 581, the word “experimental” was deleted and the sentence was rephrased. Also in the Legend of Figure 6 and Title of section 4.1.

  • Reviewer request: Please provide the diet composition in the supplementary material. This information, which otherwise ignored by many of us, is very essential for replication and reproducibility of results.

Answer to the Reviewer: Complete chemical composition of rat diet is provided in Supplementary material as Table S1. It is also indicated in the text.

  • Reviewer request: Rat “food” should be rat “diet”. Please check.

Answer to the Reviewer: Rat “food” was changed with rat “diet” in section 4.1 and in the Legend of Table 6.

  • Reviewer request: I disagree to the contention of authors regarding dosage of selenium. There have been varying opinions regarding what is actual supra-nutritional levels. Prof. Margaret Rayman has rightly said “Selenium will surely keep us guessing for some time to come”.

Answer to the Reviewer: To mitigate our statement that the selected doses of Se are supranutritive and non-toxic, we have provided an additional explanation and citation by Rayman (2020) as follows:” There have been varying opinions regarding what is optimal and what is supranutritional, non-toxic dose of Se in rats [34,35]. According to the Rayman (2020) [40], tolerance toward Se in humans depends on genes that regulate their ability to tolerate low or high doses of Se, but by now this has not been fully elucidated. Optimal dose of Se vary between populations and individuals [40]. However, the chosen dose of synthesized SeNPs was shown to be supranutritional and non-toxic, based on testing of adult male rats [34,35]”.

  • Reviewer request: Although there has been some advancement in recent years, therefore, previous recommendations (from older studies) using rodent models should be very carefully evaluated. It has been reported recently that different levels of Se have different effects on expression of selenoproteins and that also depend on organ/context in which they are being studied.

Answer to the Reviewer: We have also mentioned this statement in the Introduction section, Paragraph 3.

  • Reviewer request: For ensuring the consistency of nomenclature of selenoproteins, authors are suggested to review article of Dr. Vadim et al. 10.1074/jbc.M116.756155. We all have been strong advocate of bringing consistency in nomenclature.

Answer to the Reviewer: As recommended, we review the article Gladyshev et al (2016) to ensure consistency in the selenoproteins nomenclature. Very nice and useful article. Some citations from this article are included in the text in Introduction section, Paragraphs 1 and 3.

  • Reviewer request: Please remove “native English language speaker” from acknowledgment, as their names are enough. We better be inclusive.

Answer to the Reviewer: “native English language speaker” was removed from acknowledgment as suggested.

Once again, thank you for your time and constructive suggestions to improve quality of our manuscript.

Sincerely,

Slađan Pavlović, Ph.D.

Principal Research Fellow
